# Fusing multisensory signals across channels and time

**Swathi Anil** [ID][1,2,3], **Dan F. M. Goodman** [ID][1], **Marcus Ghosh** [ID][1,4*]

**1** Department of Electrical and Electronic Engineering, Imperial College London, London, United Kingdom, **2** Department of Neuroanatomy, Institute of Anatomy and Cell Biology, Faculty of Medicine, University of Freiburg, Freiburg, Germany, **3** Bernstein Center Freiburg, University of Freiburg, Freiburg, Germany, **4** I-X Center for AI in Science, Imperial College London, London, United Kingdom

* m.ghosh@imperial.ac.uk

**Data availability statement:** Example code for generating all tasks can be found in S1 Appendix. Full code for all tasks and models can be found at: https://github.com/swathianil/Temporal_Nonlinear_fusion.

## Abstract

Animals continuously combine information across sensory modalities and time, and use these combined signals to guide their behaviour. Picture a predator watching their prey sprint and screech through a field. To date, a range of multisensory algorithms have been proposed to model this process including linear and nonlinear fusion, which combine the inputs from multiple sensory channels via either a sum or nonlinear function. However, many multisensory algorithms treat successive observations independently, and so cannot leverage the temporal structure inherent to naturalistic stimuli. To investigate this, we introduce a novel multisensory task in which we provide the same number of task-relevant signals per trial but vary how this information is presented: from many short bursts to a few long sequences. We demonstrate that multisensory algorithms that treat different time steps as independent, perform sub-optimally on this task. However, simply augmenting these algorithms to integrate across sensory channels and short temporal windows allows them to perform surprisingly well, and comparably to fully recurrent neural networks. Overall, our work: highlights the benefits of fusing multisensory information across channels and time, shows that small increases in circuit/model complexity can lead to significant gains in performance, and provides a novel multisensory task for testing the relevance of this in biological systems.

## Author summary

We constantly detect sensory inputs, like sights and sounds, and use combinations of these signals to guide our actions. For example, by reading someone's lips we can better converse with them in a noisy environment. Several mathematical models have been proposed to describe this process. However, these models are "blind" to time. That is, following the example above, if we took the audio and visual signals from our friend and mixed them up over time; current models would not notice any difference, but we would find the result incomprehensible. Motivated by this, we introduce a new set of models

**Funding:** SA is supported by the Landesgraduiertenförderung Abschlussstipendium, issued by the Graduate Funding of the Land of Baden-Württemberg (LGFG). MG is supported by the Eric and Wendy Schmidt AI in Science Postdoctoral Fellowship, a Schmidt Sciences program. The funders had no role in study design, data collection and analysis, decision to publish, or preparation of the manuscript.

**Competing interests:** The authors have declared that no competing interests exist.

which describe how animals could fuse sensory signals across time. Surprisingly, we find that combining signals across senses and short periods of time, works as well as a more complex model.

## Key points

- We introduce a novel multisensory task in which we provide task relevant evidence via bursts of varying duration, amidst a noisy background.
- Prior multisensory algorithms perform sub-optimally on this task, as they cannot leverage temporal structure.
- However, they can perform better by integrating across sensory channels and short temporal windows.
- Surprisingly, this allows for comparable performance to fully recurrent neural networks, while using less than one tenth the number of parameters.

## Introduction

Picture a predator trying to track prey through a dense field. How should they approach this challenge? One solution would be to rely solely on either visual or auditory cues, such as sightings of, or screeches from, their prey. However, these unisensory strategies will be sub-optimal in many situations, like poor lighting conditions or noisy environments. Consequently, many animals combine information across their senses and base their decisions on these merged signals: a fundamental process termed multisensory integration [1,2].

To date, numerous algorithms have been proposed to describe *how* animals implement this process [3]. For example, *n*-look algorithms suggest that observers examine the inputs from *n* channels, but form their "multisensory" output using only one - which could be the channel with the strongest or fastest signal [4,5]. In contrast, fusion algorithms form their outputs by combining their inputs across sensory channels either linearly [6–9] or nonlinearly [3,10,11]. These algorithms can all be interpreted as instantaneous input-output mappings, or coupled to drift-diffusion models and used to explore how observers integrate multisensory evidence over time. For example, how animals determine their heading direction from visual and vestibular cues [6,8]. However, in general, these algorithms treat successive observations independently. Meaning they are unable to leverage the temporal structure inherent to naturalistic signals.

In contrast, experiments using visual [12], auditory [13] and multisensory stimuli [14–17] have demonstrated that our perception at any given moment is strongly influenced by our recent observations - a phenomenon termed serial dependence. For example, when observers are presented with a sequence of orientated Gabor's and asked to report the orientation of each; their responses will be accurate - on average across the experiment, but systematically biased by recent trials - on a trial-by-trial basis [18]. This phenomenon, has been framed as being both advantageous - as integrating information over time will improve signal-to-noise, and disadvantageous - as recent stimuli could render an instantaneous response sub-optimal [19]. However, much of this research has focused on trial-trial dependence, rather than moment-to-moment changes in dependence within a trial.

Here, we explore moment-to-moment multisensory integration in three steps. First, we introduce a novel multisensory task in which we provide the same number of task relevant stimuli per trial (in a background of noise), but vary how this information is presented: from many short bursts to a few long sequences. Next, we demonstrate that prior multisensory

algorithms perform sub-optimally on this task, though perform better by simply considering short temporal windows. Finally, we explore the more naturalistic case, in which information is structured at multiple timescales. Specifically, we sample burst lengths from a Lévy distribution. This distribution, describes animal behaviours such as foraging [20], which are composed of many short bursts (local exploitation) interspersed by occasional long flights (exploration).

## Results

### Multisensory algorithms are blind to temporal structure

We previously introduced a family of multisensory tasks in which observers must track prey using sequences of multisensory signals over time (Fig 1A) [11]. Conceptually, in these tasks, prey either hide or emit signals, which provide clues about their direction of motion, at every time step and observers must estimate their direction of motion (e.g. left or right). Practically, in each trial, prey are assigned a direction of motion. Then, at each time point, a hidden variable ($e$) takes values 0 or 1. When $e = 0$ (prey hiding), the predator observes noise (signals with no bias in either direction) in each sensory channel. When $e = 1$ (prey observable), the predator receives signals which indicate the prey's direction of motion. As such, while each time step provides little information on it's own, observers can improve their accuracy by accumulating evidence over time. However, like many prior multisensory tasks each time step is independent; meaning that consecutive sequences of signals are no more informative than those same signals spaced over time.

Here, we introduce a new multisensory task with two key properties:

1. Prey emit bursts of cues which are informative of their direction of motion. The length of these bursts is set by a parameter we term $k$. Low $k$ generates short bursts, while high $k$ generates long sequences (Fig 1B). When $k = 1$ there is no serial dependence and each time step is independent.
2. As $k$ increases we decrease the number of bursts, such that the total number of time steps where the signal is present, or *signal sparsity*, is constant across trials (on average). For example, for a trial of a given length and signal sparsity, we would provide $n$ bursts for a burst length of 4, $2n$ bursts for a burst length of 2 and $4n$ bursts for a burst length of 1.

Together, these properties generate trials in which the total duration of the signal is constant, but how this information is presented varies: from many short bursts to a few long sequences. Notably, as the time steps in this task are not independent, calculating optimal performance, in the Bayesian sense, is computationally intractable [21] and so we cannot easily compute an upper bound on performance.

We began by training and testing two algorithms (linear and nonlinear fusion *LF*, *NLF*) on this task as we varied the burst length $k$ from 1 to 8 (Fig 1C and 1D). When we allow these algorithms access to only one channel (unisensory) both perform well (Fig 2A). Though, they perform significantly better with access to both channels (multisensory) demonstrating the benefit of fusing information across channels in this task. Consistent with prior work [11] *NLF* outperformed *LF* in the multisensory case. At lower levels of signal sparsity, this difference would increase substantially [11], though here our focus is on temporal structure, not linear vs nonlinear fusion. Notably, both algorithms' accuracies decrease slightly as a function of $k$, despite the additional temporal information available, and consistent signal sparsity. The reason for this is that, while the average number of signal events remains constant across

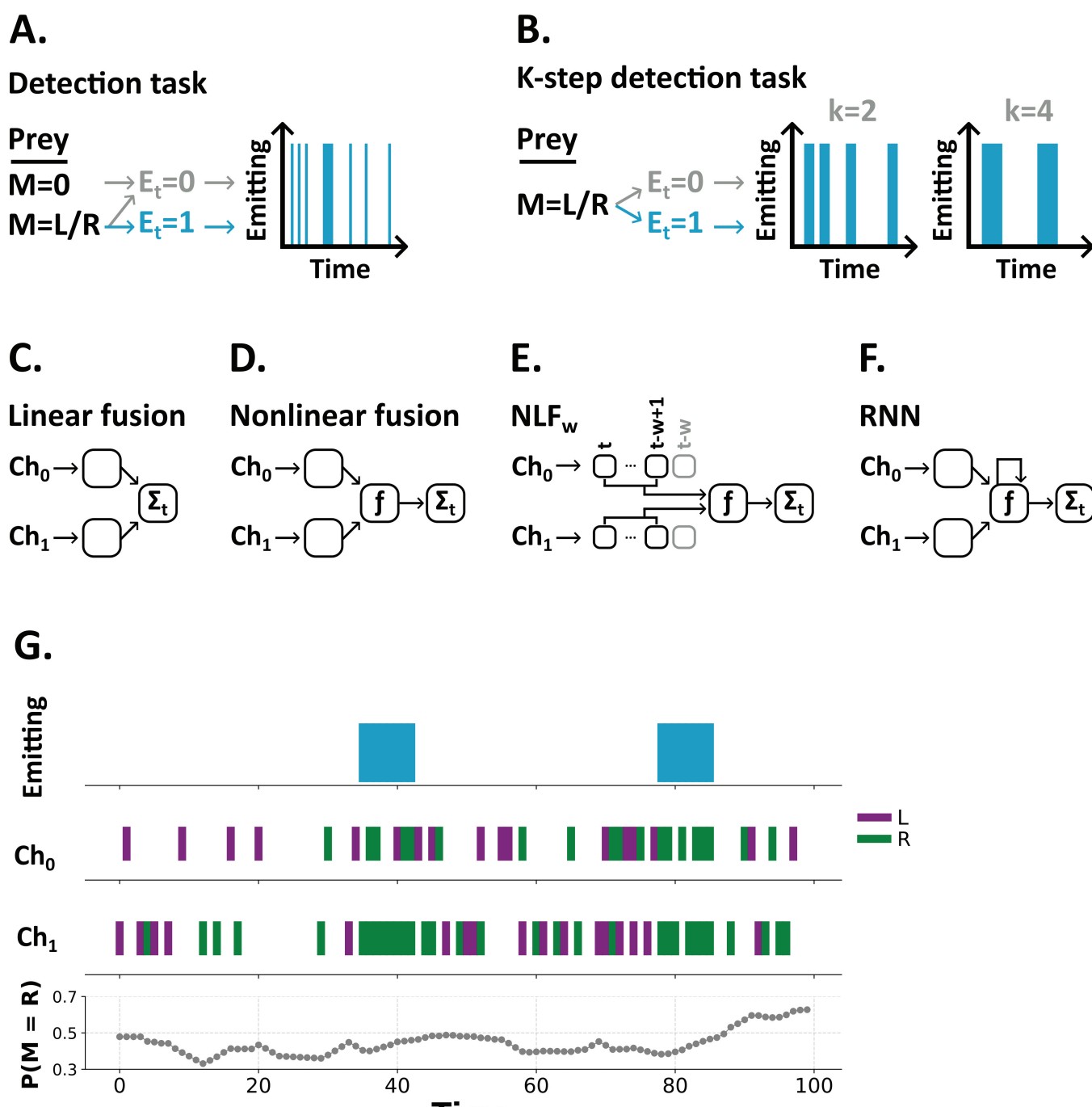

**Fig 1.** In the multisensory detection task from [11] (**A**), observers must estimate their prey's motion (left/right) from sequences of time-independent signals. Here, we introduce a variant of this task (**B**) in which, prey emit either short bursts or long sequences of signals controlled by a burst length parameter $k$. Notably, as $k$ increases we decrease the number of bursts, such that the total number of time steps where the signal is present is constant across trials (on average). For example, here when $k = 2$ we provide 4 bursts, while when $k = 4$ we provide only 2. In classical multisensory algorithms the information from independent sensory channels ($Ch_0$, $Ch_1$) is combined via a linear sum (**C**) or nonlinear function (**D**) then summed over time. To capture temporal structure we adapted the nonlinear fusion algorithm to process sliding input windows of length $w$ (**E**). We compare these models to fully recurrent neural networks (**F**). **G** An example trial showing whether or not the target was emitting a signal (top row, variable $E$), the sensory signals in the two channels (middle two rows), and the corresponding estimated probability that the target is moving right ($M = R$), as estimated by a model, in the bottom row. As evidence accumulates the model gains confidence in its prediction.

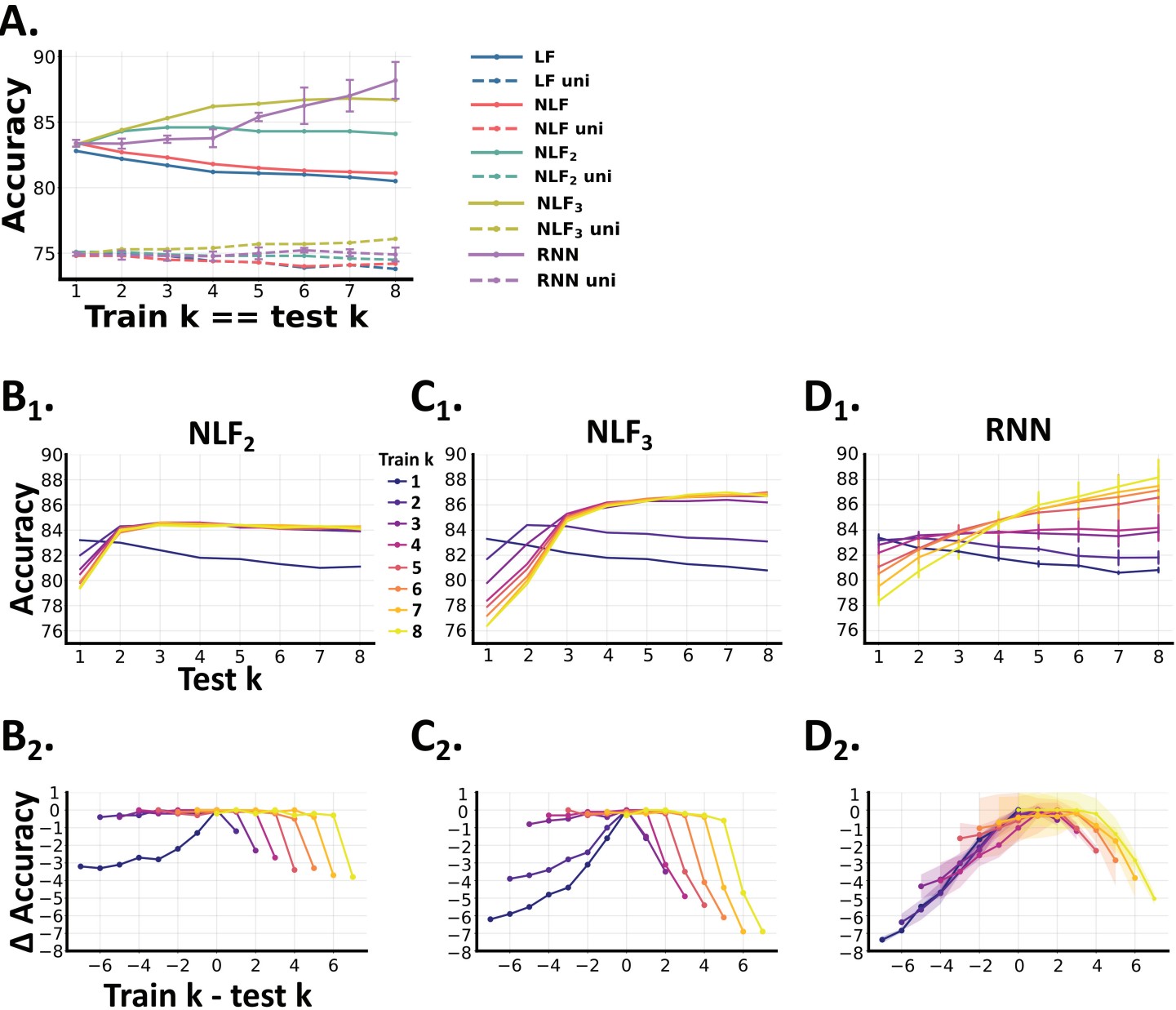

**Fig 2. A** The accuracy of each model (y-axis) when trained and tested on the same value of the signal burst length $k$ (x-axis); i.e. when tested in distribution. We trained and tested 5 *RNN* models, and plot their mean accuracy $\pm$ standard deviation. **$B_1 - D_1$** The test accuracy of each model when trained on a specific value of $k$ (colours) and tested on another (x-axis), i.e. when tested out of distribution. **$B_2 - D_2$** The accuracy of the algorithm trained on $k$ and tested on $k - \Delta k$ (where $\Delta k$ is the x-axis), relative to the best achievable accuracy by that model on that test set. More precisely, if $A(k_{train}, k_{test})$ is the accuracy when trained on $k_{train}$ and tested on $k_{test}$, then the plot shows $A(k, k - \Delta k) - \max_{k'} A(k', k - \Delta k)$. When $\Delta k < 0$, each model is being tested on longer sequences than it was trained on. When positive, models are being tested on sequences shorter than they were trained on.

$k$, the distribution of the number of signal events, in a given trial, has some variability. This is explained in more detail in S1 Appendix.

Overall, these results illustrate the intuitive point that, as often implemented, linear and nonlinear fusion across channels cannot leverage temporal structure.

## Incorporating time into multisensory algorithms

To capture temporal information, we adapted the nonlinear fusion algorithm to process sliding input windows of length $w$; a family of models we term $NLF_w$ (Fig 1E). As such, at each time step ($t$), these models combine incoming ($t$) and prior signals (from $t-w$). As the number of parameters in these models scales unfavourably ($O^{wN_c}$), we focus on $NLF_2$ and $NLF_3$. To capture dependencies over longer timescales, we also trained recurrent neural network models (*RNNs*). Our *RNN* models used nonlinear activations (ReLU) and were trained with backpropagation through time. Overall, our models have the following numbers of learnable parameters: *LF* - 6, *NLF* - 9, $NLF_2$ - 81, $NLF_3$ - 729, *RNN* - 10,903 (Table 1). Notably, a linear algorithm with a sliding window is equivalent to one without (as the order of operations does not matter), so there is no need to test these models.

**In distribution.** We, first considered how well these models performed *in distribution*. That is, when each model is trained and tested on bursts of the same length ($k$): train on $k = 2$, test on $k = 2$, etc. (Fig 2A).

When $k = 1$, $NLF_{2/3}$ and the *RNN* models all perform equivalently to *NLF*, as each time step is independent and there is no additional temporal structure to detect. However, as $k$ increases these models outperform *LF* and *NLF* as they are able to leverage the additional temporal information available. Though, *how* each model's accuracy varies with $k$ differs. $NLF_{2/3}$ increase their performance then plateau. While, *RNN* performance is flat then rises. As such, while, the *RNNs* excel at detecting longer sequences, the simpler $NLF_w$ models are better at detecting shorter bursts and surprisingly good at longer sequences, even when those sequences are longer than their window length (or memory). This is likely due to the fact that, even when $k$ is high, short bursts are sufficiently informative of the prey's direction of motion; as they are unlikely to occur in the incorrect direction. Moreover, when we trained smaller *RNNs*, with an equivalent number of trainable parameters to $NLF_3$, we found that they performed far worse, suggesting that $NLF_w$ may be a better model per parameter (S2 Fig). In S3 Fig, we visualise the learned parameters for $LF, NLF, NLF_2, NLF_3$ models as a function of $k$.

Notably, unisensory versions of these models performed worse than their multisensory equivalents (Fig 2A) and showed less difference in their performance as a function of $k$ (S4 Fig).

Together, these results demonstrate the benefits of fusing information over channels and time. However, in naturalistic settings, predators must perform well not only in response to motion patterns they have experienced, but also, in novel, unseen situations.

**Table 1.** A table noting: each model's parameter scaling, each model's number of learnable parameters here, and how each model combines information across sensory channels (linear or nonlinear) and time. Linear and nonlinear fusion (*LF* and *NLF*), treat each timestep independently. $NLF_{2/3}$ fuse information across short temporal windows. *RNNs* combine incoming signals with their prior hidden states. For the parameter scaling functions: $O$ - the number of possible observations, $N_c$ - the number of sensory channels, $w$ - the temporal integration window. For our *RNN* models the number of parameters scales as: $(N_I \times N_H) + (N_H \times N_H) + (N_b \times N_H) + (N_H \times N_O) + N_O$, where $N_I, N_H, N_O$ denote the number of input, hidden and output units and $N_b$ is the number of bias parameters per hidden unit.

| Model | Parameter scaling | Parameters here | Fusion across channels | Fusion across time |
|---|---|---|---|---|
| *LF* | $O \times N_c$ | 6 | Linear | Independent |
| *NLF* | $O^{N_c}$ | 9 | Nonlinear | Independent |
| $NLF_2$ | $O^{wN_c}$ | 81 | Nonlinear | 2 timestep window |
| $NLF_3$ | $O^{wN_c}$ | 729 | Nonlinear | 3 timestep window |
| *RNN* | see caption | 10,903 | Nonlinear | Prior hidden states |

**Generalisation.** We next considered how well these models *generalise*. That is how well they perform when they are trained on one burst length ($k$) and tested on another (Fig 2B, 2C and 2D).

When fit on $k = 1$ and tested on $k>1$, all three model's accuracies decrease slightly as a function of $k$ (dark blue lines, Fig $2B_1$, $2C_1$ and $2D_1$). This reflects the fact that while these models have the capacity to detect sequences, there is no benefit learning to do so when your training data has no temporal structure ($k = 1$). As such, they perform equivalently to *NLF* (Fig 2A). Similarly, when $NLF_3$ is trained on $k = 2$ it only performs as well as $NLF_2$; and perhaps even slightly worse (Fig $2C_1$), as there is no benefit learning to detect longer sequences.

In contrast, as these models learn from longer sequences, past $w$ in the case of $NLF_{2/3}$, all three models generalise reasonably well; the maximum difference in accuracy we observe is less than 8% (Fig $2B_2$, $2C_2$ and $2D_2$). Though, again, we observe a notable difference between the $NLF_w$ and *RNN* models. Specifically, both *NLF* models generalise better when tested on longer sequences than they were trained on, and less well to shorter sequences. In contrast, the *RNN* generalises better to shorter rather than longer sequences (Fig $2B_2$, $2C_2$ and $2D_2$).

Overall these results, demonstrate that all three models generalise well when tested out of distribution. However, these scenarios are still unrealistic, in the sense that the prey always emit bursts of similar length.

## Capturing multi-timescale structure

To add further realism to our task, we next considered a variant in which, within each trial, prey emit bursts of varied length; drawn from either a uniform or Lévy distribution. The latter is a heavy-tailed distribution, which we chose to explore, as it describes animal behaviours like foraging [20] which are composed of many short bursts, broken by occasional long flights (which constitute the heavy tail).

We found that all three models ($NLF_2$, $NLF_3$ and *RNNs*) performed well when trained and tested on burst lengths drawn from either uniform or Lévy distributions. Furthermore, models trained on our fixed length tasks generalised well to these mixed length tasks (Fig 3A, 3B and 3C).

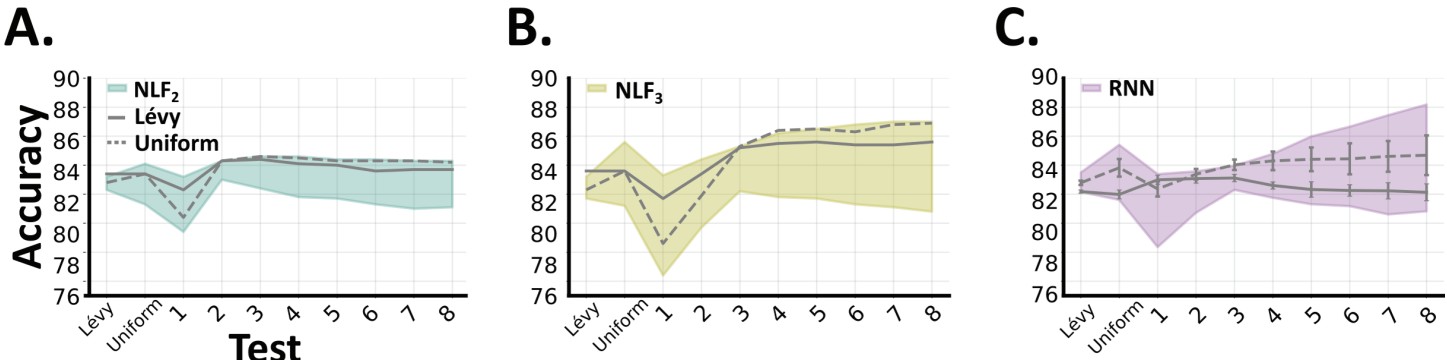

**Fig 3.** Model performance (test accuracy, y-axis) on mixed (uniform and Lévy) and fixed burst length tasks (x-axis). **A** Nonlinear fusion with window length 2 ($NLF_2$), **B** Nonlinear fusion with window length 3 ($NLF_3$), **C** Recurrent neural networks (*RNN*). In each sub-panel, the solid/dashed gray lines show how well the model performs when it is trained on mixed burst lengths (Lévy/uniform), and then tested on mixed or fixed burst length tasks (across the x-axis). Each shaded region shows the min-max span across all models trained on fixed burst lengths (from $k = 1 - 8$). We calculated this by: training models for each value of $k$, testing them on all bursts (mixed and fixed), then selecting the best and worst model per scenario.

However, $NLF_w$ models trained on mixed distributions tended to generalise better than those trained on fixed distributions (Fig 3A and 3B). Though, the same was not true for the *RNNs* (Fig 3C). For example, at $k = 8$, the $NLF_w$ models trained on mixed distributions outperformed those trained on fixed distributions (Fig 3A and 3B). While, the *RNNs* trained on fixed distributions outperformed their counterparts trained on mixed distributions (Fig 3C). This suggests that learning to detect a mixture of short bursts, in the case of $NLF_{2/3}$, yields a parsimonious strategy that generalises well to longer sequences. While, with more resources (i.e. parameters) the *RNNs* can learn more specialised strategies (for each value of $k$).

Together these results demonstrate that in a more complex situation, where prey emit signals in multiple channels and at multiple time-scales, all three models perform reasonably well. However, across settings (testing in and out of distribution, on both fixed and mixed length signals), $NLF_3$ often performs best or close to best (Fig 4). Underscoring the benefit of fusing information, not only over channels, but also over short temporal windows.

## Discussion

To date, many psychophysical tasks have been used to explore how animals combine information across their senses. In classical multisensory tasks, each sensory modality (or channel) provides evidence about an underlying target *independently* [6–9,22]. As such, the optimal solution to these tasks is to *linearly* fuse (or integrate) evidence across channels. Though, in the case when channels are co-dependent the optimal solution is to *nonlinearly* fuse evidence across channels [11]. This case seems likely to arise in natural conditions; consider the relation between lip movements and sounds, for example. However, in both of these cases evidence should be fused *linearly across time*. This is because within these tasks, each time step is independent, meaning the time points within each trial could be shuffled without changing the results.

Here, we explored another scenario that seems likely to arise in natural conditions, where the evidence at any given moment depends on recent moments: like the bursts of sensory signals prey darting from cover to cover would emit. To do so, we adapted a task from [11] to make the target (prey) emit bursts of varied length; thereby introducing a sequential time

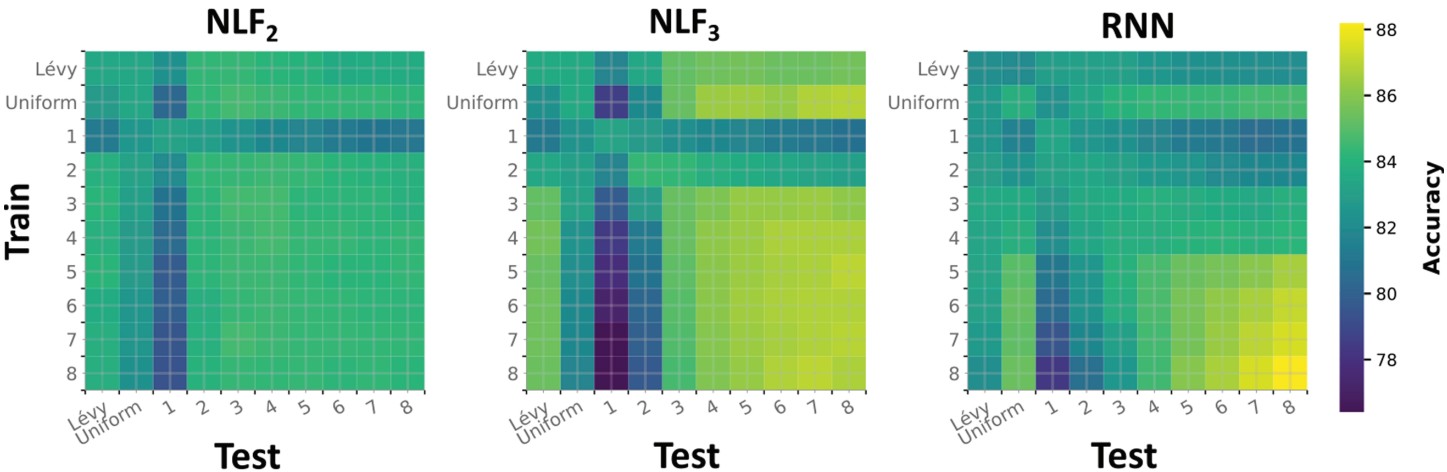

**Fig 4. Performance of $NLF_2$, $NLF_3$ and $RNN$ across various training and testing conditions; in distribution (on the diagonal) and out of distribution (off diagonal).** We train and test on 10 different distributions: Lévy flights, uniformly distributed burst lengths, and fixed burst lengths ($k = 1, \ldots, 8$).

dependence. From the point of view of estimating the target value (the predator's perspective), this means that if you have recently seen evidence suggesting that the target is currently emitting a signal, you may wish to increase the relative weighting of the current evidence.

This intuition suggested two simple models which could perform this computation, and plausibly be implemented by neural circuits. The first combines information across channels and short temporal windows of a fixed length; a family of models we term nonlinear fusion over $w$ time steps, or $NLF_w$. The second are recurrent neural networks ($RNNs$) which combine the current evidence from each channel with their prior hidden states, and so can capture structure at multiple timescales. Notably, the Bayes optimal solution to this problem involves a combinatorial computation over all time steps [21], which is computationally intractable and unlikely to be implemented by neural circuits.

As expected, both models ($NLF_w$ and $RNNs$) were able to leverage temporal structure, and outperform time-independent algorithms (linear and nonlinear fusion). Though, across settings (in and out of distribution, and on both fixed and mixed length signals) $NLF_{2/3}$ performed well, and were surprisingly comparable to the $RNNs$ despite having less than one tenth the number of parameters. This suggests that in these tasks, short bursts are sufficiently informative of the target's motion and there is little benefit to detecting longer sequences. In practice, this is akin to change detection, and would also allow for faster reaction times than waiting to observe longer sequences.

So, which is a more plausible model of how neural circuits integrate information over channels and time? $NLF_w$ could be implemented by a range of simple mechanisms. For example, a multisensory neuron receiving $w$ inputs per channel, each offset with a different temporal delay would fuse information across channels and time. In contrast, the $RNN$ model requires a population of densely connected units, and hence a higher energetic cost. For the tasks we explore here, this cost does not seem merited by the increase in performance. However, in tasks with even more detailed temporal structure, for example, complex multistep, multisensory sequences, it seems likely that a recurrent network would outperform the simpler model.

In conclusion, our results demonstrate the benefits of combining multisensory information across short temporal windows ($NLF_w$) or prior states ($RNNs$). Either of which could easily be implemented by neural circuits. More broadly, our results underscore the importance of exploring more complex multisensory tasks, and highlights the fact that, despite their apparent complexity, these tasks are often solvable with simple, biologically plausible extensions to existing models.

## Materials and methods

In short, we build on the detection task from [11] - in which observers must estimate the heading direction of a target from sequences of information in $n$ channels; which represent different sensory modalities or independent sources of information from the same modality. In this task, at some (unknown) time points, the target emits signals and the sensory observations provide information about the target direction, while at other times, the sensor receives noise. In [11] we showed that this task structure requires nonlinear fusion across channels to solve optimally, but since each time step is independent, only linear fusion across time steps was needed. Here we introduce limited temporal dependency in the signals by having signals switch on at unknown times but then remain on for a number of steps (where this number is either a constant or chosen from a random distribution). It is not feasible to compute the optimal classifier in this case, and so we investigated two classifiers with different types of short-term memory, one based on a linear sum of nonlinear functions operating on a window

that slides across all time steps, and one based on a recurrent neural network. Below, we detail these tasks and inference models.

## Tasks

We start by sampling a random direction $M = \pm 1$ with equal probability. The task is to estimate $M$.

We define a sequence of binary valued random variables $E_t$ to indicate whether the signal is emitting a signal at time $t$ (where $t = 1, \dots, n$). When $E_t = 1$ the distribution of the observations $X_{it}$ in channel $i$ follow a signal distribution (giving the correct value of $M$ with probability $p_c$, incorrect value with probability $p_i$ and a neutral value (0) with probability $1 - p_c - p_i$). When $E_t = 0$, $X_{it}$ follows a noise distribution taking the correct/incorrect values of $M$ with equal probability $(1 - p_n)/2$ and the neutral value with probability $p_n$.

To generate the sequence $E_t$, we sample a generating sequence $G_t$ of signal start times, so that whenever $G_t = 1$ the signal $E_t = 1$ and will stay 1 for some period $L_t$ (which can either be a constant or be drawn from a distribution). Finally, to ensure that the task is solvable, we filter out all cases where $E_t = 0$ for all $t$. We set the probability that $G_t = 1$ so that the fraction of the time that $E_t = 1$ is equal to a value $p_e$ that we choose.

**General task structure.** In more detail, the task variables are related by the following graphical model:

$$
\begin{array}{ccc}
\{L_t\}_t & & M \\
\downarrow & & \downarrow \\
\{G_t\}_t \longrightarrow \{E_t\}_t & \longrightarrow & \{X_{it}\}_{i,t}
\end{array}
\tag{1}
$$

where

- $M \in \{-1, 1\}$ is the target direction, with each of the two values being equally likely.
- $G_t \in \{0, 1\}$ are the start times for emission periods, taking value 1 with probability $p_g$ to indicate the start of an emission period.
- $L_t \in \{1, \dots, L_{\max}\}$ is the length of the emission period starting at $t$, and follows a different distribution for different tasks.
- $E_t = \max_{t-L_t < s \leq t} G_s \in \{0, 1\}$ is whether or not the target is emitting a signal at time $t$, and has deterministic dependence on $L_t$ and $\{G_s\}_{t-L_{\max} < s \leq t}$.
- $X_{it} \in \{-1, 0, 1\}$ is the signal received in channel $i \in \{1, \dots, N_c\}$ at time $t$. It's distribution depends on $M$ and $E_t$ as described below.
- The $G_t$ are independent variables except that if $E_1 = E_2 = \cdots = E_n = 0$ then values are resampled, introducing a small time dependence that we ignore in the analysis.

The distribution of $X_{it}$ depends on whether or not there is a signal being emitted or not. If not, it follows a noise distribution that is independent of $M$:

$$
(X_{it}|E_t = 0) = \begin{cases} \pm 1 & \text{with probability } (1 - p_n)/2 \\ 0 & \text{with probability } p_n \end{cases}
\tag{2}
$$

If a signal is being emitted, then it depends on $M$ as follows:

$$(X_{it}|E_t = 1) = \begin{cases} M & \text{with probability } p_c \\ -M & \text{with probability } p_i \\ 0 & \text{with probability } 1 - p_c - p_i \end{cases} \tag{3}$$

**Task variants.** In the time-dependent detection task with on-time $k$ ($\text{Det}_k$), we have that $L_t = k$ for all $t$. In the Lévy flight task, we have that:

$$P(L_t = \ell) = \frac{1/\ell^2}{\sum_{m=1}^{L_{\max}} 1/m^2} \tag{4}$$

## Normalising for signal sparsity

In order to make performance comparable between the different task variants, we normalise certain parameters so that the average amount of useful information (ignoring time) is the same across tasks. Specifically, we choose $p_g$ so that the expected fraction of the time that a signal is being emitted $\mathbb{E}[E_t]$ is equal to a value $p_e$ that we choose. We outline the calculation of this normalisation for the $\text{Det}_k$ and Lévy flight tasks below.

**Detection task with $k$ timesteps ($Det_k$).** In this task, we sample values $G_t \in \{0, 1\}$ with $P(G_t = 1) = p_g$ independently. We compute $E_t = \max_{t-k < s \leq t} G_s$ and then resample if all $E_1 = \cdots = E_n = 0$. Note that to compute $E_1$ we need to have a value for $G_{1-k+1}$ (and onwards to $G_n$). We start by computing $P(E_t = 1)$ without the resampling procedure, and then compute the effect of resampling.

Without resampling,

$$P(E_t = 0) = P(G_{t-k+1} = \cdots = G_t = 0).$$

Each of these values $G_{t-k+1}$ to $G_t$ are independent, and therefore we can write

$$P(E_t = 0) = \prod_{s=t-k+1}^{t} P(G_s = 0)$$
$$= (1 - p_g)^k.$$

We can then compute

$$P(E_t = 1) = 1 - P(E_t = 0) = 1 - (1 - p_g)^k.$$

Now write $F$ for the event that $E_1 = \cdots = E_n = 0$ that we filter out. We want to compute the fraction of the time $E_t = 1$ when this event does not take place, $P(E_t = 1 \mid \neg F)$. To calculate this, we note that either $F$ does or does not take place, so computing the total probability over both these two possibilities we get

$$P(E_t = 1) = P(E_t = 1 \mid F)P(F) + P(E_t = 1 \mid \neg F)P(\neg F).$$

We computed $P(E_t = 1)$ above and by the definition of $F$ we know that $P(E_t = 1 \mid F) = 0$, and therefore we get that

$$P(E_t = 1 \mid \neg F) = \frac{P(E_t = 1)}{P(\neg F)}.$$

It remains to calculate

$$P(\neg F) = 1 - P(F) = 1 - P(G_{1-k+1} = \cdots = G_n = 0) = 1 - (1 - p_g)^{n+k-1}.$$

This gives us

$$p_e = P(E_t = 1 \mid \neg F) = \frac{1 - (1 - p_g)^k}{1 - (1 - p_g)^{n+k-1}}.$$

We can then numerically invert this to get the correct value for $p_g$ given a desired value $p_e$. Note that by taking limits as $p_g \to 0$, the smallest $p_e$ achievable is $k/(n + k - 1)$.

**Lévy flight.** The calculation for the Lévy flight task is similar to the $\text{Det}_k$ task but a little more involved.

To simplify notation later, write

$$q_\ell = P(L_t = \ell) = \frac{1/\ell^2}{\sum_{m=1}^{L_{\max}} 1/m^2}$$

and

$$Q_\ell = P(L_t < \ell) = \sum_{\ell' < \ell} q_\ell.$$

To compute $P(E_t = 0)$ we need to consider various possibilities depending on the on lengths $L_s$ for $s \leq t$. For $E_t = 0$, the first thing that needs to be true is that $G_t = 0$ (because if $G_t = 1$ then automatically $E_t = 1$). The next condition we need to be true is that either $G_{t-1} = 0$ or $G_{t-1} = 1$ but $L_{t-1} = 1$ meaning that the on period generated by $G_{t-1} = 1$ was only of length 1 and therefore did not cause $E_t = 1$. And so on until we have gone back to the maximum number of previous steps $L_{\max}$ that could cause $E_t = 1$. Each of these events depend on $G_s$ and $L_s$ for a different value of $s$, and are therefore independent, so

$$P(E_t = 0) = \prod_{\ell=1}^{L_{\max}} P(G_{t-\ell+1} = 0 \text{ or } (G_{t-\ell+1} = 1 \text{ and } L_{t-\ell+1} \leq \ell - 1))$$

$$= \prod_{\ell=1}^{L_{\max}} P(G_{t-\ell+1} = 0) + P(G_{t-\ell+1} = 1)P(L_{t-\ell+1} \leq \ell - 1)$$

$$= \prod_{\ell=1}^{L_{\max}} 1 - p_g + p_g Q_\ell$$

This gives us $P(E_t = 1)$ without filtering, and to compute $P(E_t = 1 \mid \neg F)$ we need to compute $P(F)$. We break this event down into independent events $F_\ell$ defined as $G_t = 0$ for $t = 1 - \ell + 1, \ldots, n$ whenever $L_t = \ell$. In other words, there can be no length-1 on periods that cause an $E_t = 1$ (event $F_1$), no length-2 on periods (event $F_2$) and so on. These are each independent events and therefore

$$P(F) = \prod_{\ell=1}^{L_{\max}} P(F_\ell) = \prod_{\ell=1}^{L_{\max}} (1 - P(G_t = 1, L_t = \ell))^{n+\ell-1} = \prod_{\ell=1}^{L_{\max}} (1 - p_g q_\ell)^{n+\ell-1}.$$

Putting it together,

$$p_e = P(E_t = 1 \mid \neg F) = \frac{1 - \prod_{\ell=1}^{L_{\max}} 1 - p_g + p_g Q_\ell}{1 - \prod_{\ell=1}^{L_{\max}} (1 - p_g q_\ell)^{n+\ell-1}}.$$

As before, this can be inverted numerically to compute the $p_g$ value that gives a desired $p_e$ value.

## Task parameters

For all tasks we used the following parameters: $p_c = 0.45$, $p_i = 0.01$, $p_n = 0.33$ and $p_e = 0.04$ which results in the following values of $p_g$ for each value of $k$ (from 1-8): 0.039, 0.02, 0.013, 0.01, 0.008, 0.006, 0.005, 0.004, and values of 0.022 / 0.011 for our mixed $k$ tasks (Lévy/uniform).

## Inference

The task is to estimate $M$ from $X_{it}$ with unknown hidden variables $G_t$, $E_t$ and $L_t$. It is straightforward to write down the optimal maximum a posteriori (MAP) estimator using Bayes' theorem. We simply want to choose $m$ that maximises $P(M = m \mid \mathbf{X}) = P(\mathbf{X} \mid M = m)P(M = m)/P(\mathbf{X})$, and given $P(M = m) = 1/2$ and $P(\mathbf{X})$ doesn't depend on $m$, we just need to maximise $P(\mathbf{X} \mid M = m)$. We can marginalise over the hidden variables to get

$$P(\mathbf{X} \mid M = m) = \sum_{\mathbf{g}, \boldsymbol{\ell}} P(\mathbf{X} \mid M = m, \mathbf{G} = \mathbf{g}, \mathbf{L} = \boldsymbol{\ell}) = \sum_{\mathbf{g}, \boldsymbol{\ell}} \prod_t P(\mathbf{X}_t \mid M = m, \mathbf{G} = \mathbf{g}, \mathbf{L} = \boldsymbol{\ell}).$$

However, given that the time steps are not independent, and the sum is over $(2 + L_{\max})^n$ terms, it is hard to simplify this any further in a way that could be easily computable by the brain.

Instead, we consider two types of estimators with different types of *short term memory*.

**NLF$_w$.** The first type, $w$-step nonlinear fusion (or NLF$_w$) computes, for each time step, a likelihood based on the previous $w$ time steps (which can be implemented as a table look up over all the possible observations $X_{it}$ over $w$ time steps), and then sums this over all time steps. In more detail, we enumerate every possible observation $X_{it}$ for $t = 1, \ldots, w$ (of which there are $3^{wN_c}$ possibilities). For example, for $N_c = 2$, $w = 2$, the 81 observations are $X_{it} = \begin{pmatrix} -1 & -1 \\ -1 & -1 \end{pmatrix}, \begin{pmatrix} -1 & -1 \\ -1 & 0 \end{pmatrix}, \ldots, \begin{pmatrix} 1 & 1 \\ 1 & 1 \end{pmatrix}$. For every trial, we then then count how often each of these observations occurs for every window $t = [1 \cdots w], [2 \cdots w + 1], \cdots [n - w + 1 \cdots n]$, and make these counts into a feature vector $\mathbf{F} \in \mathbb{N}^{3^{wN_c}}$. For example, if $\mathbf{X} = \begin{pmatrix} -1 & -1 & -1 & 1 \\ -1 & -1 & -1 & 0 \end{pmatrix}$ then the count for the observations $\begin{pmatrix} -1 & -1 \\ -1 & -1 \end{pmatrix}$ is 2, and the count for the observation $\begin{pmatrix} -1 & 1 \\ -1 & 0 \end{pmatrix}$ is 1, and all other observations have a count of 0. We then train a logistic regression classifier to estimate $M$ from $\mathbf{F}$ using scikit-learn [23]. The parameters of the linear part of the model can be interpreted as the log-likelihoods of each possible observation in a window, and using counts is equivalent to summing these over all windows.

**RNN.** The second type of estimator, *RNN*, uses a recurrent neural network with one hidden layer and 100 hidden units, fed at each time step with inputs $X_{it}$ and its previous state $\mathbf{h}_{t-1}$, and the output of this network is treated as a likelihood. In more detail, write $\mathbf{h}_t$ for the activity of the hidden layer at time step $t$, and $\mathbf{o}_t$ for the activity of the output layer.

$$\mathbf{h}_t = \text{ReLU}(\mathbf{X}_t W_{ih}^\mathsf{T} + \mathbf{h}_{t-1} W_{hh}^\mathsf{T} + \mathbf{b}^h) \tag{5}$$

$$\mathbf{o}_t = \text{ReLU}(\mathbf{h}_t W_{ho}^\mathsf{T} + \mathbf{b}^o) \tag{6}$$

for nonlinear activation functions $f$, $g$, weight matrices $W_{ih}$, $W_{hh}$ and $W_{ho}$, and biases $\mathbf{b}^h$ and $\mathbf{b}^o$. The rectified linear unit function is defined as:

$$\text{ReLU}(x) = \begin{cases} x & \text{if } x \geq 0 \\ 0 & \text{if } x < 0. \end{cases}$$

The hidden state $\mathbf{h}_0$ is initialized to zero, and outputs are summed across time steps to produce the final prediction $\hat{m} = \text{argmax}_m \sum_t o_{mt}$. The network is trained with a standard cross-entropy loss on the output layer using PyTorch [24] and the Adam optimiser [25] with learning rate of $10^{-6}$.

## Code

Example code for generating all tasks can be found in S1 Appendix. Full code for all tasks and models can be found at: https://github.com/swathianil/Temporal_Nonlinear_fusion.

## Supporting information

**S1 Appendix. Why does task performance decrease with burst length? And task code.** (PDF)

**S1 Fig. Why does task performance decrease with burst length?** Explanation of why accuracy decreases for methods that do not use temporal information (*LF*, *NLF*) as $k$ increases. See S1 Appendix. (TIFF)

**S2 Fig. Performance of algorithms compared to a smaller *RNN*.** The accuracy of each model ($y$-axis) when trained and tested on the same value of the signal burst length $k$ ($x$-axis); i.e. when tested in distribution. For the *RNN* models error bars show standard deviation over 5 separate train/test runs. For comparison, we include the performance of a scaled down *RNN* (*RNN$_{small}$*), with 22 hidden units and 663 trainable parameters—a comparable number of parameters to *NLF$_3$* (729 parameters). (TIFF)

**S3 Fig. Parameters of trained algorithms.** Learned parameter weights. Each subplot shows the weights learned by a single model (*LF*, *NLF*, *NLF$_2$* and *NLF$_3$*). Each x-axis shows the value of $k$ used to train the model. Each y-axis shows possible observations / features. For example $\left(\begin{smallmatrix} L & R \\ R & R \end{smallmatrix}\right)$; sorted by their total corresponding parameter value (summed across values of $k$). Intuitively, ambiguous signals tend to be assigned low/zero weights. For example, $\left(\begin{smallmatrix} L & R \\ R & L \end{smallmatrix}\right)$. While, unambigous signals, tend to be assigned larger weights. For example, $\left(\begin{smallmatrix} L & L \\ L & L \end{smallmatrix}\right)$. And *NLF$_w$* seems to learn different patterns when $k<w$ vs when $k \geq w$. Though we did not quantify these observations. (TIFF)

**S4 Fig. Comparison of multi- and unisensory task performances.** Scatter plot showing the performance of algorithms in unisensory ($x$ axis) and multisensory ($y$ axis) tasks. Accuracies are shown relative to the accuracy of the corresponding performance of the nonlinear fusion (*NLF*) algorithm. Lighter colours correspond to lower values of the burst length $k$, and darker values to higher values. (TIFF)

## Acknowledgements

We thank members of the Neural Reckoning group at Imperial for their helpful input.

## Author contributions

**Conceptualization:** Dan F. M. Goodman, Marcus Ghosh.

**Data curation:** Swathi Anil.

**Formal analysis:** Swathi Anil, Dan F. M. Goodman, Marcus Ghosh.

**Investigation:** Swathi Anil.

**Methodology:** Swathi Anil, Dan F. M. Goodman, Marcus Ghosh.

**Project administration:** Dan F. M. Goodman, Marcus Ghosh.

**Resources:** Dan F. M. Goodman, Marcus Ghosh.

**Supervision:** Dan F. M. Goodman, Marcus Ghosh.

**Visualization:** Swathi Anil.

**Writing – original draft:** Swathi Anil, Dan F. M. Goodman, Marcus Ghosh.

**Writing – review & editing:** Swathi Anil, Dan F. M. Goodman, Marcus Ghosh.

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
