## [Decision Letter · Decision Letter 0]

12 Feb 2025

PCOMPBIOL-D-24-02205

Fusing multisensory signals across channels and time

PLOS Computational Biology

Dear Dr. Goodman,

Thank you for submitting your manuscript to PLOS Computational Biology. After careful consideration, we feel that it has merit but does not fully meet PLOS Computational Biology's publication criteria as it currently stands. Therefore, we invite you to submit a revised version of the manuscript that addresses the points raised during the review process.

Please submit your revised manuscript within 60 days Apr 14 2025 11:59PM. If you will need more time than this to complete your revisions, please reply to this message or contact the journal office at ploscompbiol@plos.org. Please include the following items when submitting your revised manuscript:

We look forward to receiving your revised manuscript.

Kind regards,

Tianming Yang

Academic Editor

PLOS Computational Biology

Andrea E. Martin

Section Editor

PLOS Computational Biology

**Journal Requirements:**

At this stage, the following Authors/Authors require contributions: Swathi Anil, Marcus Ghosh, and Dan F. M. Goodman. Please ensure that the full contributions of each author are acknowledged in the "Add/Edit/Remove Authors" section of our submission form.

5) We notice that your supplementary information is included in the manuscript file. Please remove them and upload them with the file type 'Supporting Information'. Please ensure that each Supporting Information file has a legend listed in the manuscript after the references list.

**Reviewers' comments:**

Reviewer's Responses to Questions

Reviewer #1: Main findings

The authors describe an algorithm that combines information across “channels” (i.e., different modalities) and time to solve a new multisensory discrimination task. This work builds on their previously published task and algorithm (Ghosh et al., 2024), which focused on nonlinear fusion across channels only. In this new task, bits of sensory evidence are not temporally independent anymore but occur in bursts, and algorithms that treat each timestep independently perform sub-optimally. The authors propose here to nonlinearly fuse signals over short periods of time, and show that this modified model performs just as well as RNNs, with much less parameters.

By making their task temporally rich and thus more realistic, the authors provide an interesting framework to test the performance of various models. However, I found the current logic of the paper difficult to follow, especially as a biologist, and unless there are misunderstandings on my side the conceptual improvements over their previous work could be better described. Overall, it would really help if the authors could provide more detailed explanations and figures, and comparisons with the previous literature.

Major coomments

- I found the current framing of the problem in the context of multisensory integration quite misleading. As far as I understood, the multisensory aspect of the task does not seem to play a role in the results described. Couldn’t the paper be written with a single sensory channel and a temporal window? To clarify this point (both in the overall scope, the results and their explanation), it would be helpful to start with only one modality and put the emphasis on the effect of temporal integration only. Multichannel integration could then be studied in a later figure if needed.

Also, isn’t the new model equivalent to the previous one but with channels 2, 3, etc. being a delayed version of the first channel? I would find it helpful if the authors could clarify this point from the beginning.

- It would also be very useful to provide more information and explanation in the figures and main text. Without the previous paper at hand, it is very difficult to follow what is the task, and what are the models, and makes it challenging to build an intuition of how things work. The figures directly jump from a schematic description of the model to a summary of their accuracy. Additional panels would help. In particular, showing example trials with the actual signals being emitted, and the behavior of the models over the course of the trial, would help clarify how the different models integrate the sensory evidence over time. This could for example help understand why LF and NLF’s performance decreases with k (which I think I get but not sure).

Also, what do the parameters of the NLF look like for different values of trained k? The authors could provide a direct explanation of why the models fail to generalize when trained on k=1, based on these parameters. (comparing NLF to NLF2 should still be tractable for example)

Minor comments

It may be trivial but I think it would be useful to mention what a linear fusion algorithm that has a w timestep window would be/do? Would it be equivalent to the linear fusion algorithm without a window (because it accumulates over time)?

Paragraph 3.2.1: “even when those sequences are longer than their window length (or memory)” – I couldn’t get an intuition of why, it would be helpful if the authors could unpack it a bit?

Figure 2: I am not sure I understand the correspondence between B1-D1 and B2-D2 -- e.g. NLF-3 trained on k=1, in C1 the max difference in accuracy is ~3, and in C2 it’s 6. Aren’t the dark blue lines supposed to be the same?

To allow a fairer comparison of parameters numbers, it could be interesting to test smaller sizes of RNNs, and see when the RNN’s performance starts to break.

I find Figure 4 quite hard to read (e.g., what is gray?), and was wondering if 3 heatmaps (one per model) side by side with the same color scale wouldn’t be easier?

Reviewer #2: The paper presents a model that accounts for multisensory integration and time. The models are tested using a novel task which is motivated by the example of hunting a prey. The paper presents five models, two with temporal independence (Linear Fusion, LF; and Non-Linear Fusion NLF), two which account for time (NLF two time steps, NL2 and three time steps, NLF3) and an RNN as a comparison. Overall I think the paper shows some very interesting findings. My comments are really focused on clarifications and further details about the models and the results.

Introduction

The introduction is concise and readable.

Two behavioural papers that do consider the trial by trial impact on behaviour that might be of interest are Wozny & Shams 2011, Shaw et al. 2020. This does raise an interesting debate about whether the benefit seen in a multisensory trial is partially due to a repetition of signal.

The use of the Lèvy distribution is interesting and could be mentioned in the intro to telegraph it being used later in the methods.

Results

I appreciate that the format of PLoS papers has the methods at the end but some more detail about the models in the results would be nice, like number of parameters is 3^{wNc} and the RNN uses ReLUs.

In Figure 2 that caption states accuracy and standard deviation of each model but the only model with visible error bars is the RNN, why is this?

Some more detail about the structure of the Lèvy distribution would be nice, just a sentence.

Are the results in Figure 3 train and tested on the same k?

Also, in Figure 3 is the highlighted area trained on mixed distributions while the dashed lined are trained on only levy or only uniform. And is the highlighted area representing the standard deviation.

Figure 4 would benefit from a bit more explanation in how to interpret the results. One possible way of helping the reader for example would be to box the lower right 4x4 grid for Test 5-8 and Train 5-8 where the RNN is “winning”.

Discussion

Like the introduction, the discussion is readable and concise.

Methods

While it is important to show that the equations can generalise to w time-steps and N_c channels. As the paper focuses on 2 and 3 time steps and two channels it might be nice to show an example of these simpler versions, only a suggestion.

For the simulations in the paper what values or values of the task parameters (such as pg and pe used)

Just to a clarification could the burst be in one or both channels Xit and if so did the authors consider training a single channel version or a version with burst only on a single channel, to simulate a unisensory task.

5.2.1 should have the subheading Detection task with k timesteps (Detk)

How long did it take to train the models?

The RNN could have its own subheading.

References

Wozny, D. R., & Shams, L. (2011). Recalibration of auditory space following milliseconds of cross-modal discrepancy. Journal of Neuroscience, 31(12), 4607-4612.

Shaw, L. H., Freedman, E. G., Crosse, M. J., Nicholas, E., Chen, A. M., Braiman, M. S., ... & Foxe, J. J. (2020). Operating in a multisensory context: Assessing the interplay between multisensory reaction time facilitation and inter-sensory task-switching effects. Neuroscience, 436, 122-135.

**Have the authors made all data and (if applicable) computational code underlying the findings in their manuscript fully available?**

Reviewer #1: Yes

Reviewer #2: Yes

PLOS authors have the option to publish the peer review history of their article (what does this mean?). If published, this will include your full peer review and any attached files.

Reviewer #1: No

Reviewer #2: **Yes: **John Butler

**Figure resubmission:**
---

## [Decision Letter · Decision Letter 1]

8 May 2025

Dear Dr Ghosh,

We are pleased to inform you that your manuscript 'Fusing multisensory signals across channels and time' has been provisionally accepted for publication in PLOS Computational Biology.

Best regards,

Tianming Yang

Academic Editor

PLOS Computational Biology

Andrea E. Martin

Section Editor

PLOS Computational Biology

Reviewer's Responses to Questions

**Comments to the Authors:**

Reviewer #1: I thank the authors for addressing all of my comments. I just two have minor clarification remarks.

My suggestion on unisensory models was not to compare unisensory vs. multisensory models in this multisensory task, but just to start with a simpler unisensory task to study models with and without fusion in time (which is the new thing in this paper), before going to the more complex multisensory task and models. The authors do not need to change this if they don’t feel like it, but I think it would help clarify the first paragraphs of the results (they already have the figure so it’s just rewording).

Fig 1G: could the authors mention which model was used to plot the timecourse?

Congratulations on the paper!

Reviewer #2: The authors have addressed my comments

**Have the authors made all data and (if applicable) computational code underlying the findings in their manuscript fully available?**

Reviewer #1: Yes

Reviewer #2: Yes

PLOS authors have the option to publish the peer review history of their article (what does this mean?). If published, this will include your full peer review and any attached files.

Reviewer #1: No

Reviewer #2: **Yes: **John Butler

---

## [Editor Report · Acceptance letter]

PCOMPBIOL-D-24-02205R1

Fusing multisensory signals across channels and time

Dear Dr Ghosh,

I am pleased to inform you that your manuscript has been formally accepted for publication in PLOS Computational Biology. Your manuscript is now with our production department and you will be notified of the publication date in due course.

With kind regards,

Lilla Horvath
